# Synergistic Application of Multiple Machine Learning Algorithms and Hyperparameter Optimization Strategies for Net Ecosystem Productivity Prediction in Southeast Asia

Chaoqing Huang [1,2,3], Bin Chen [4], Chuanzhun Sun [5], Yuan Wang [6], Junye Zhang [2,3], Huan Yang [2,3], Shengbiao Wu [4], Peiyue Tu [2,3], MinhThu Nguyen [7], Song Hong [2,3] and Chao He [1,*]

1 College of Resources and Environment, Yangtze University, Wuhan 434023, China; hcqwhu@126.com
2 School of Resource and Environmental Science, Wuhan University, Wuhan 430079, China; tupeiyue@foxmail.com (P.T.); songhongpku@126.com (S.H.)
3 Key Laboratory of Geographic Information System, Ministry of Education, Wuhan University, Wuhan 430079, China
4 Future Urbanity & Sustainable Environment (FUSE) Lab, Division of Landscape Architecture, Department of Architecture, Faculty of Architecture, The University of Hong Kong, Hong Kong, China
5 School of Public Management, South China Agricultural University, Guangzhou 510642, China
6 School of Geography and Environment, Jiangxi Normal University, Nanchang 330022, China
7 Vietnam Institute of Meteorology Hydrology and Climate Change, Ministry of Natural Resources and Environment, Hanoi City 100803, Vietnam
* Correspondence: hechao@yangtzeu.edu.cn

**Abstract:** The spatiotemporal patterns and shifts of net ecosystem productivity (NEP) play a pivotal role in ecological conservation and addressing climate change. For example, by quantifying the NEP information within ecosystems, we can achieve the protection and restoration of natural ecological balance. Monitoring the changes in NEP enables a more profound understanding and prediction of ecosystem alterations caused by global warming, thereby providing a scientific basis for formulating policies aimed at mitigating and adapting to climate change. The accurate prediction of NEP sheds light on the ecosystem's response to climatic variations and aids in formulating targeted carbon sequestration policies. While traditional ecological process models provide a comprehensive approach to predicting NEP, they often require extensive experimental and empirical data, increasing research costs. In contrast, machine-learning models offer a cost-effective alternative for NEP prediction; however, the delicate balance in algorithm selection and hyperparameter tuning is frequently overlooked. In our quest for the optimal prediction model, we examined a combination of four mainstream machine-learning algorithms with four hyperparameter-optimization techniques. Our analysis identified that the backpropagation neural network combined with Bayesian optimization yielded the best performance, with an $R^2$ of 0.68 and an MSE of 1.43. Additionally, deep-learning models showcased promising potential in NEP prediction. Selecting appropriate algorithms and executing precise hyperparameter-optimization strategies are crucial for enhancing the accuracy of NEP predictions. This approach not only improves model performance but also provides us with new tools for a deeper understanding of and response to ecosystem changes induced by climate change.

**Keywords:** machine-learning algorithms; hyperparameter optimization; remote sensing; Southeast Asia; net ecosystem productivity

## 1. Introduction

Net ecosystem productivity (NEP) refers to the residual part of net primary productivity after subtracting the consumption of photosynthetic products by heterotrophic respiration [1]. As an indicator of ecosystem productivity, NEP plays a pivotal role in the surface carbon cycle, offering an intuitive reflection of vegetation productivity in natural environments. It serves as a critical metric in characterizing the land ecosystem's response



to climate change and directly impacts the global carbon cycle and climate stability [2,3]. A positive NEP suggests that the ecosystem assimilates more carbon from the atmosphere than it releases, acting as a carbon sink, vital in offsetting carbon emissions from industrialization and urbanization. In contrast, a negative NEP might result from ecosystem degradation, wildfires, or changes in land use, turning it into a carbon source. Therefore, accurately predicting the spatiotemporal variations of land NEP is crucial for devising effective climate change mitigation strategies and ensuring the continuous contribution of ecosystems to regional and global carbon sinks [4,5].

For a long time, terrestrial NEP has predominantly been measured based on experimental methods [6]. With the progression of computer technology and numerical modeling, the methodologies for predicting NEP transformed as vegetation ecological process models became widely adopted [7–10]. In recent years, with the emergence of big data and machine-learning technologies, there have been breakthroughs in the prediction methods for NEP. Utilizing machine-learning algorithms, such as random forests and deep learning, researchers can extract information about NEP from extensive remote-sensing, meteorological, and ecological datasets [11–13]; but given the limitations in data-driven model structures, there can be a loss in the detailed representation of certain critical ecological processes and internal ecosystem variations, leading to discrepancies in simulation results [14–16]. These discrepancies could be between the model outputs and actual scenarios or among different model outcomes. Secondly, while machine-learning approaches offer new possibilities for NEP prediction and excel in numerical simulation accuracy, they introduce their challenges, notably the scale effect. When predicting NEP on a larger scale using machine learning, there is a risk of overlooking or smoothing out phenological and environmental variations significant on smaller scales [17,18]. On a global scale, machine-learning models might be influenced by inconsistent spatial intensities of observational data, resulting in a deviation from actual scenarios [19]. Furthermore, some scientists have already applied traditional machine-learning methods, such as random forests, to predict NEP on a global scale. However, variations in machine-learning algorithms and tuning strategies can produce differing results. Specifically, there is limited research on applying deep-learning algorithms in this area. Therefore, constructing and comparing deep-learning models with traditional machine-learning models for iterative optimization might be the most anticipated outcome in regionally precise NEP prediction [20–23]. For regions rich in ecosystem productivity, constructing regional multi-machine-learning algorithm models based on appropriate observation points and long-term data, comparing various hyperparameter-optimization combinations, and further optimizing model performance could lead to an optimal model for accurately inverting regional NEP.

The objectives of our research are: (1) Utilize machine-learning algorithms, specifically random forest, support vector regression, backpropagation neural network, and convolutional neural network, fine-tuned with hyperparameter-optimization strategies like random search, grid search, Bayesian optimization, and genetic algorithms to process long-term NEP observations and remote-sensing data. This is to accurately predict the annual NEP in Southeast Asia, linking algorithm selection and optimization directly to the predictive accuracy of our model. (2) Determine the most effective model for NEP prediction in Southeast Asia by analyzing the performance of each algorithm–strategy combination, ensuring that the chosen model captures the unique phenological signatures of the region's ecosystems. (3) Validate our model by comparing its NEP predictions against international studies, establishing its effectiveness in capturing regional ecosystem productivity. These steps are devised to fulfill the hypothesis that different machine-learning approaches will yield varied NEP predictions in Southeast Asia, from which the optimal solution will be systematically derived.

## 2. Data and Methods

### 2.1. Study Area Overview

The study area is Southeast Asia (92°E–140°E, 10°S–28°26′N), comprising the Indo-China Peninsula and the Malay Archipelago (Figure 1). The northern part of the Indochina Peninsula is mountainous, while the southern part is relatively flat. The Malay Peninsula and the Malay Archipelago are predominantly hilly and mountainous. To the north, Southeast Asia borders China; to the east, it faces the Pacific Ocean; to the south, it overlooks Australia across the sea; and to the west, it gazes upon the Indian Ocean. It comprises Brunei, Cambodia, Indonesia, Laos, Malaysia, Myanmar, the Philippines, Singapore, Thailand, and Vietnam, covering approximately 4.49 million square kilometers. Most Southeast Asian countries are situated near the equator, resulting in relatively stable annual temperatures and abundant precipitation, predominantly characterized by tropical rainforest and monsoon climates [24].

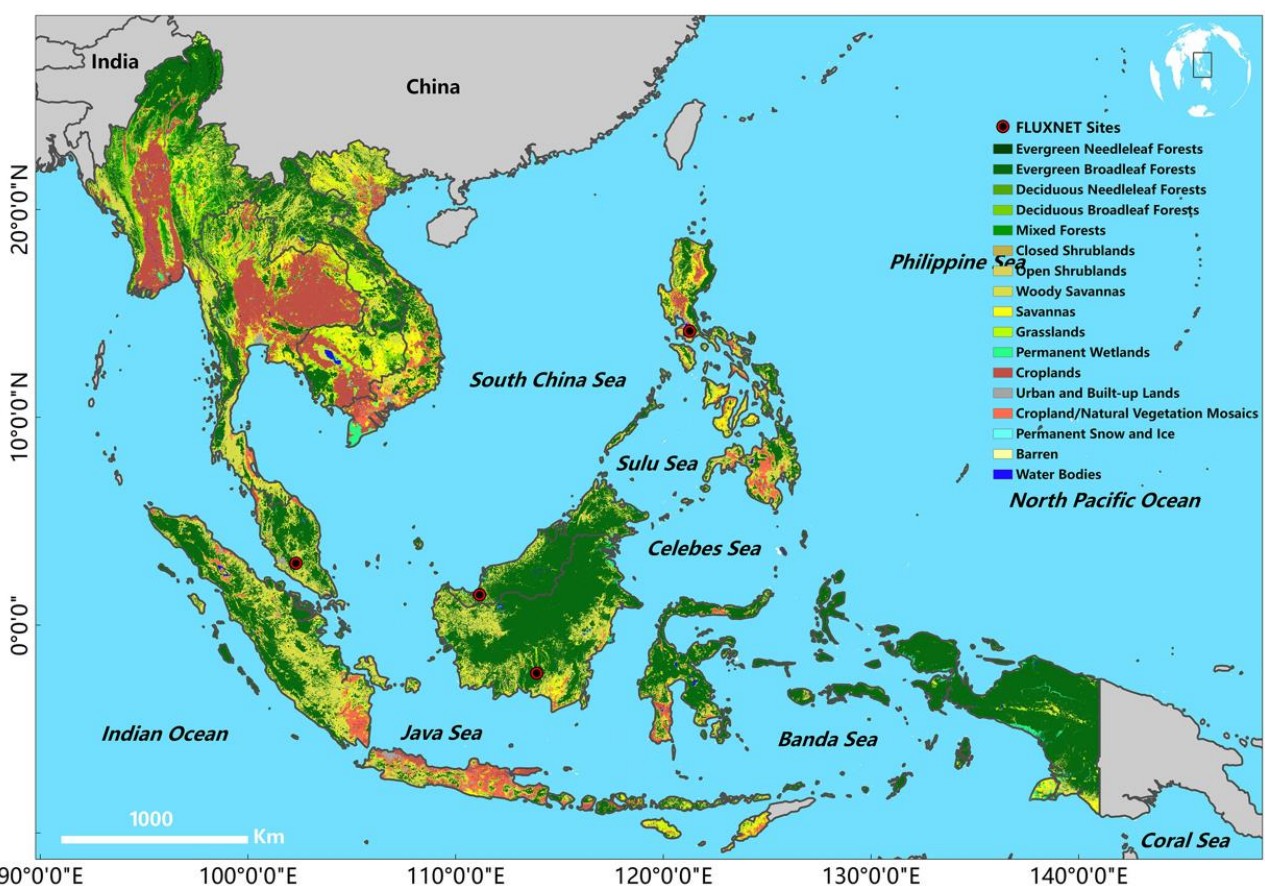

**Figure 1.** Land cover and FLUXNET Sites in South east Asia.

### 2.2. Data and Pre-Processing

In this study, the prediction of NEP was conducted based on the relationship 'NEP = −NEE' [25,26]. NEE is a comprehensive indicator encompassing the net carbon contribution of all biological activities and soil respiration in an ecosystem to the atmosphere [27,28]. Positive values indicate that the ecosystem acts as a carbon source, while negative values signify it as a carbon sink. We set NEE as the target feature, with the feature variables being sensible heat flux (H), latent heat flux (LE), longwave radiation (LW), shortwave radiation (SW), vapor pressure deficit (VPD), atmospheric pressure (PA), air temperature (TA), precipitation (P), wind speed (WS), and normalized difference vegetation index (NDVI). These features were chosen as they play pivotal roles in the carbon cycling processes between ecosystems and the atmosphere [29–32]. Accordingly, we collated daily

data from four FLUXNET observation sites in Southeast Asia (Figure 1) spanning from 2003 to 2016, totaling ten types of observational data, which included one target feature and nine feature variables, detailed in Table 1, resulting in 4,333 records. After handling missing values, invalid values (−9999), and extreme values (i.e., values outside the 1% and 99% quantiles for each observational dataset of the site), 4181 valid records remained. As the vegetation index plays a crucial role in vegetation productivity, we collected NDVI values corresponding to the observation site locations and periods using GEE, establishing it as the tenth feature variable.

**Table 1.** Dataset.

| Parameter | Data Type | Original Spatial Resolution (m) | Data Type |
|---|---|---|---|
| NEE, H, LE, SW, LW, VPD, PA, TA, P, WS | CSV | / | FLUXNET2015 Dataset [1] |
| NDVI | CSV | / | MCD43A4.061 [2] |
| NEE, H, LE, SW, LW, VPD, PA, TA, P, WS | tif | 11,132 | ERA5 Monthly Aggregates [3] |
| NDVI | tif | 11,132 | MCD43A4.061 [2] |
| NEP | tif | / | NIES [4] |
| NEP | tif | / | National Earth System Science Data Center National Science and Technology Infrastructure of China [5] |

[1] https://fluxnet.org/data/fluxnet2015-dataset/ (accessed on 22 August 2023), [2] https://developers.google.com/earth-engine/datasets/catalog/MODIS_061_MCD43A4 (accessed on 13 September 2023), [3] https://developers.google.com/earth-engine/datasets/catalog/ECMWF_ERA5_LAND_MONTHLY_AGGR (accessed on 13 September 2023), [4] https://www.nies.go.jp/doi/10.17595/20200227.001-e.html (accessed on 15 September 2023), [5] http://www.geodata.cn/data (accessed on 15 September 2023).

Given the significant ecological changes in Southeast Asia since the onset of the new millennium, we opted to predict the NEP for this area from 2001 to 2020 [33,34]. We batch-acquired monthly remote-sensing image data for the ten variables over 20 years (2001–2020) via the GEE platform. The resolution was standardized to 11,132 m, with geographical coordinates set to WGS1984. Additionally, we collated annual NEP raster products produced by NIES and GEODA for comparative validation (Table 1).

### 2.3. Methods

In studying the carbon exchange of ecosystems, we possess a continuous sequence of 4181 observational records, which capture critical natural condition indicators including NEE, H, LE, LW, SW, VPD, PA, TA, P, WS, and NDVI. Given the dataset's spatiotemporal richness and inherent complexity, we meticulously selected machine-learning models tailored to each aspect of our objectives. To predict the annual NEP with high precision, we employ random forest (RF) for its robustness in noisy, high-volume data and its ensemble approach that aggregates multiple decision trees to improve prediction accuracy. Support vector regression (SVR) is utilized for its ability to capture complex, nonlinear relationships which is crucial for modeling the intricate interactions present within our meteorological data, aiding in the identification of the most optimal model for NEP prediction. A backpropagation neural network (BPNN) is integrated to approximate the multifaceted functional relationships between climatic indicators and carbon exchange, offering nuanced insights into the interplays of variables. This choice is driven by the need to understand the detailed interactions within our dataset to ensure the selection of the best model. Furthermore, the convolutional neural network (CNN) is introduced to harness its strength in processing time-sequential data, aligning with the temporal continuity of our observations, and autonomously extracting spatiotemporal features, ensuring the model's predictions are robust and comprehensive. Each algorithm's deployment is strategically chosen to address specific objectives, ensuring that our methodology is not just a collection of tools

but a suite of purpose-driven analyses that provide clarity and depth to our understanding of ecosystem carbon exchange. The comprehensive research framework of this study is depicted in Appendix A.

### 2.3.1. Random Forest Algorithm (RF)

As a contemporary ensemble learning method within machine learning, RF has been extensively deployed for various issues, particularly regression problems. Its foundational principle hinges on constructing and amalgamating the estimative prowess of multiple decision trees to approximate intricate data distributions [35,36]. In regression scenarios, the operational mechanism of random forests primarily manifests in two facets: initially, bootstrapping techniques are employed to extract numerous sample subsets from the original data, with each subgroup independently training an individual decision tree. Subsequently, only a random fraction of features are computed during each node split to infuse further diversity and circumvent the overfitting inherent in solitary decision trees. This stochastic feature selection strategy ensures each tree possesses a unique construction modality, amplifying the model's generalization capacity. When conducting estimations, the random forest averages the estimative values from all decision trees, culminating in a more robust result [37].

### 2.3.2. Support Vector Regression (SVR)

SVR, a regression form of the support vector machine, aspires to identify a hyperplane that maximizes the margin between data points and the decision boundary, thereby furnishing a reliable solution for regression problems [38,39]. In traditional regression techniques, the objective is to minimize the discrepancy between estimated and actual values. Conversely, SVR intends to ensure the error does not surpass a predetermined threshold $\varepsilon$ while concurrently striving to minimize the model's complexity. This approach can be perceived as striking a balance between estimation error and model intricacy. At the heart of SVR is mapping data to a higher-dimensional feature space, wherein a linear optimal hyperplane is sought. To realize this aim, SVR harnesses the kernel trick to implicitly compute the dot product in the feature space within the original data domain, sidestepping computational intricacies [40]. The choice of kernel function is pivotal to SVR's efficacy, with prevalent kernels encompassing linear, polynomial, and radial basis function kernels. Through its unique strategy of maximizing margins and the kernel trick, SVR proffers an efficient, dependable solution to regression challenges, boasting commendable generalization capabilities.

### 2.3.3. Backpropagation Neural Network (BPNN)

BPNN represents a deep-learning algorithm, offering a robust framework for tackling intricate nonlinear relationships. At its core, it estimates through forward propagation and updates weights via backpropagation to minimize the discrepancy between estimated and actual values [41]. In the context of regression, BP neural networks endeavor to identify a continuous function mapping, deriving continuous outputs from input feature spaces. Each layer within the network encompasses several neurons that undergo nonlinear transformations through activation functions, such as Sigmoid, ReLU, and others, thereby bolstering the model's expressive capacity. The retrograde weight updates in the model are accomplished via the gradient descent method, which computes the partial derivatives of the loss function for each weight, adjusting the weights subsequently to diminish errors [42]. Against deep learning, BPNN can be equipped with multiple hidden layers, thereby discerning advanced patterns and features intrinsic to the data [43]. Training profound BPNNs may necessitate supplementary techniques, such as early stopping, regularization, dropout, and batch normalization, to avert overfitting and expedite convergence.

### 2.3.4. Convolutional Neural Network (CNN)

CNN is a cornerstone architecture in deep learning, initially conceived for handling data with grid-like structures, such as image classification. CNN captures local and global

data features by stacking multiple convolutional layers. Each convolutional layer acts as a feature extractor, capable of detecting specific patterns in the data. As the network delves deeper, these detections become increasingly abstract and intricate, facilitating profound data recognition [44,45]. However, CNN's prowess is not confined to classification tasks; it also manifests substantial potential in regression problems. Owing to its depth and parameter-sharing properties, CNN adeptly encapsulates and represents intricate data distributions, bolstering continuous value estimation. For regression tasks, the output layer of a CNN is typically designed to produce one or multiple continuous values rather than classification labels, with the loss function pivoting from classification errors to estimation errors, such as mean squared error [46–48]. While CNNs have achieved monumental success in image recognition, their applicability and potential in regression challenges also merit keen attention and further exploration.

### 2.3.5. Hyperparameter-Optimization Strategy

Hyperparameter tuning is a pivotal step to ensure the optimal performance of a model. In this study, we incorporate four hyperparameter-optimization strategies, namely random search (RS), grid search (GS), Bayesian optimization (BO), and genetic algorithms (GA). RS is a strategy that randomly selects hyperparameters. It operates independently of previously evaluated hyperparameter combinations, randomly choosing a new set of hyperparameters during each iteration. The merit of this approach lies in its simplicity and straightforward implementation, with the ability to sidestep local optima, potentially pinpointing favorable hyperparameter combinations in a relatively brief span [49].

In contrast, GS adopts a more structured approach, conducting an exhaustive search within a predefined hyperparameter space and evaluating each conceivable combination [50]. While potentially time-consuming, it ensures the identification of the optimal hyperparameter set within the stipulated range.

Conversely, BO employs a more intricate strategy, leveraging probabilistic models to predict which hyperparameter combinations might yield superior results, prioritizing searches in these realms [50]. Its primary advantage is its ability to intelligently pinpoint hyperparameter combinations for evaluation, securing optimal solutions in fewer iterations.

Inspired by biological evolution, GA deploys strategies like selection, crossover, and mutation to navigate the hyperparameter space [50]. Characterized by their ability to maintain a population of hyperparameter combinations and gravitate towards optimal solutions based on performance, genetic algorithms are particularly apt for complex, non-continuous, and ill-defined hyperparameter realms.

## 3. Results

### 3.1. Application of Multi-Algorithm Predictions for Annual NEP in Southeast Asia

#### 3.1.1. Results of Random Forest Algorithm

In this study, the RF algorithm was combined with four hyperparameter-optimization strategies, each determined through cross-validation to achieve an optimal model performance. For ease of comparison, the hyperparameters, including the number of trees (Trees), maximum depth of the tree (Depth), minimum number of samples required to split an internal node (Min split), and minimum number of samples required to be at a leaf node (Min leaf), as well as the hyperparameter-optimization strategies' number of iterations (N_ITER), cross validation (CV), and population size (PS) are presented in Table 2. It can be observed that different optimization strategies yield various hyperparameter combinations, reflecting the characteristics and preferences of each strategy when searching the hyperparameter space.

Figure 2 comprehensively shows the validation comparison between the observed and model-predicted values obtained using the four hyperparameter-optimization strategies: RS, GS, BO, and GA. We observed that the predictions from the RS, GS, and BO strategies are consistent with the actual values (with $R^2$ values ranging between 0.68 and 0.7 and MSE values between 1.43 and 1.47). However, the results from the GA strategy are more dispersed,

with an $R^2$ value of 0.22 and MSE of 3.5. After comparing the performance of $R^2$ and MSE, we believe that the model results obtained using the RS strategy are the most reliable.

**Table 2.** Optimal hyperparameter results of RF algorithm combined with four hyperparameter-optimization strategies.

| Optimization Strategy | Trees | Depth | Min split | Min leaf | N_ITER | CV | PS |
|---|---|---|---|---|---|---|---|
| RS | 200 | 20 | 5 | 1 | 100 | 3 | / |
| GS | 500 | 40 | 10 | 4 | / | 3 | / |
| BO | 122 | 12 | 2 | 1 | 100 | 3 | / |
| GA | 100 | None | 2 | 1 | 50 | / | 20 |

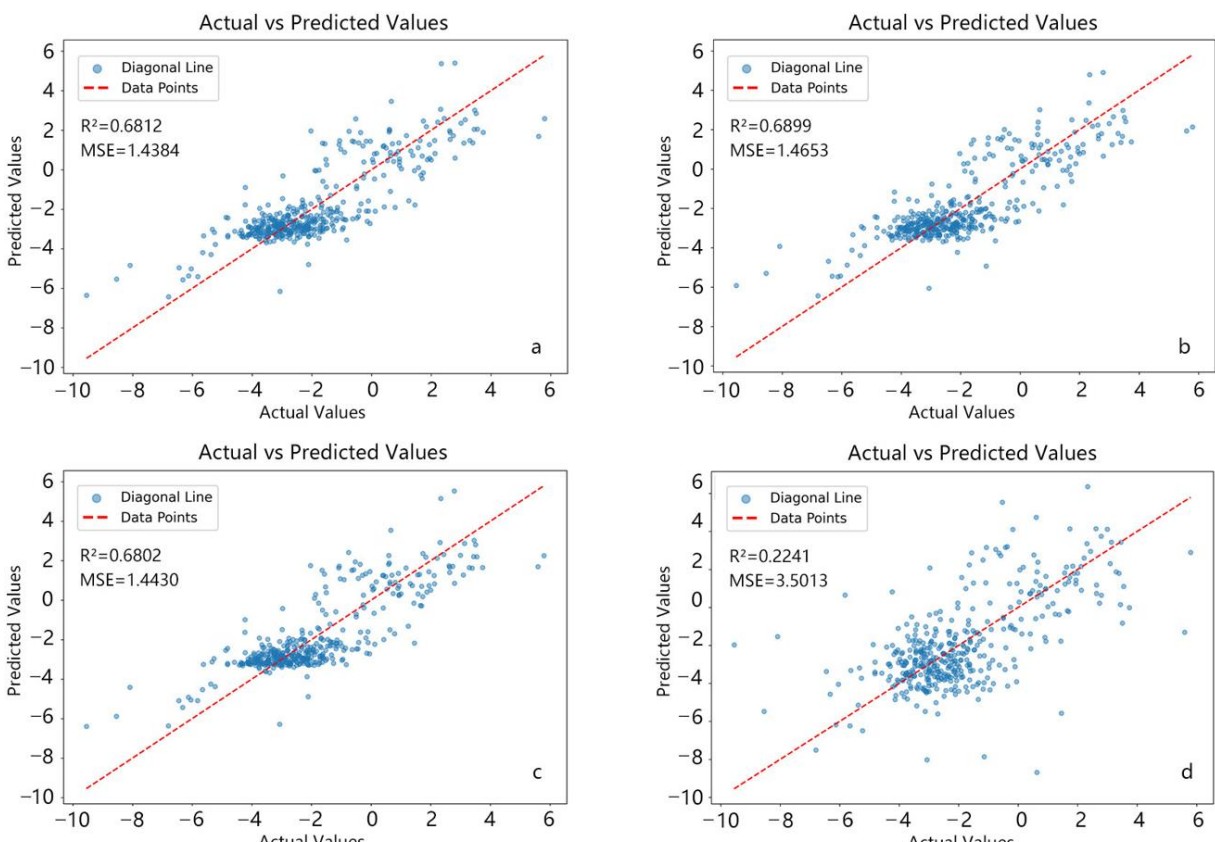

**Figure 2.** Comparison of the RF algorithm combined with different hyperparameter-optimization strategies, RS (**a**), GS (**b**), BO (**c**), GA (**d**).

### 3.1.2. Results of the Support Vector Regression Algorithm

To delve deeper into the performance of the SVR algorithm combined with the four hyperparameter-optimization strategies, we employed the cross-validation method, ensuring that each optimization strategy could achieve its optimal performance. To visually present the model's key hyperparameters, we consolidated the kernel function (Kernel), epsilon-insensitive loss (epsilon), and regularization parameter (C) along with the optimization strategy hyperparameters: number of iterations (N_ITER), cross validation (CV), and population size (PS), and presented them in tabular form (Table 3). Among them, the kernel function describes the mapping mechanism of data in high-dimensional space; the epsilon-insensitive loss defines a permissible error range; errors beyond this range are only considered, while C, as a regularization parameter, determines the model's error tolerance. The various optimization strategies demonstrated distinct characteristics and tendencies during their search process in the parameter space.

**Table 3.** Optimal hyperparameter results of SVR algorithm combined with four hyperparameter-optimization strategies.

| Optimization Strategy | Kernel | Epsilon | C | N_ITER | CV | PS |
|---|---|---|---|---|---|---|
| RS | RBF | 1.0 | 10.0 | 100 | 3 | / |
| GS | RBF | 1.0 | 10.0 | / | 3 | / |
| BO | RBF | $1 \times 10^{-6}$ | 0.2208 | 100 | 3 | / |
| GA | RBF | 1.0 | 10.0 | 50 | / | 10 |

Figure 3 provides a detailed representation of the observed versus predicted values validated by the four hyperparameter-optimization strategies, RS, GS, BO, and GA, as applied in the support vector regression model. The RS, GS, and GA strategies demonstrated an outstanding prediction accuracy, yielding an $R^2$ of 0.6816 and an MSE of 1.4368. In contrast, the BO strategy lagged slightly, with its $R^2$ at 0.5946 and an MSE of 1.8296. Considering both the $R^2$ and MSE metrics, we believe that within the framework of the SVR algorithm, the models derived from the RS, GS, and GA strategies consistently showcase a high predictive capability, accurately capturing variations in the target features. In contrast, the BO strategy appears slightly less effective.

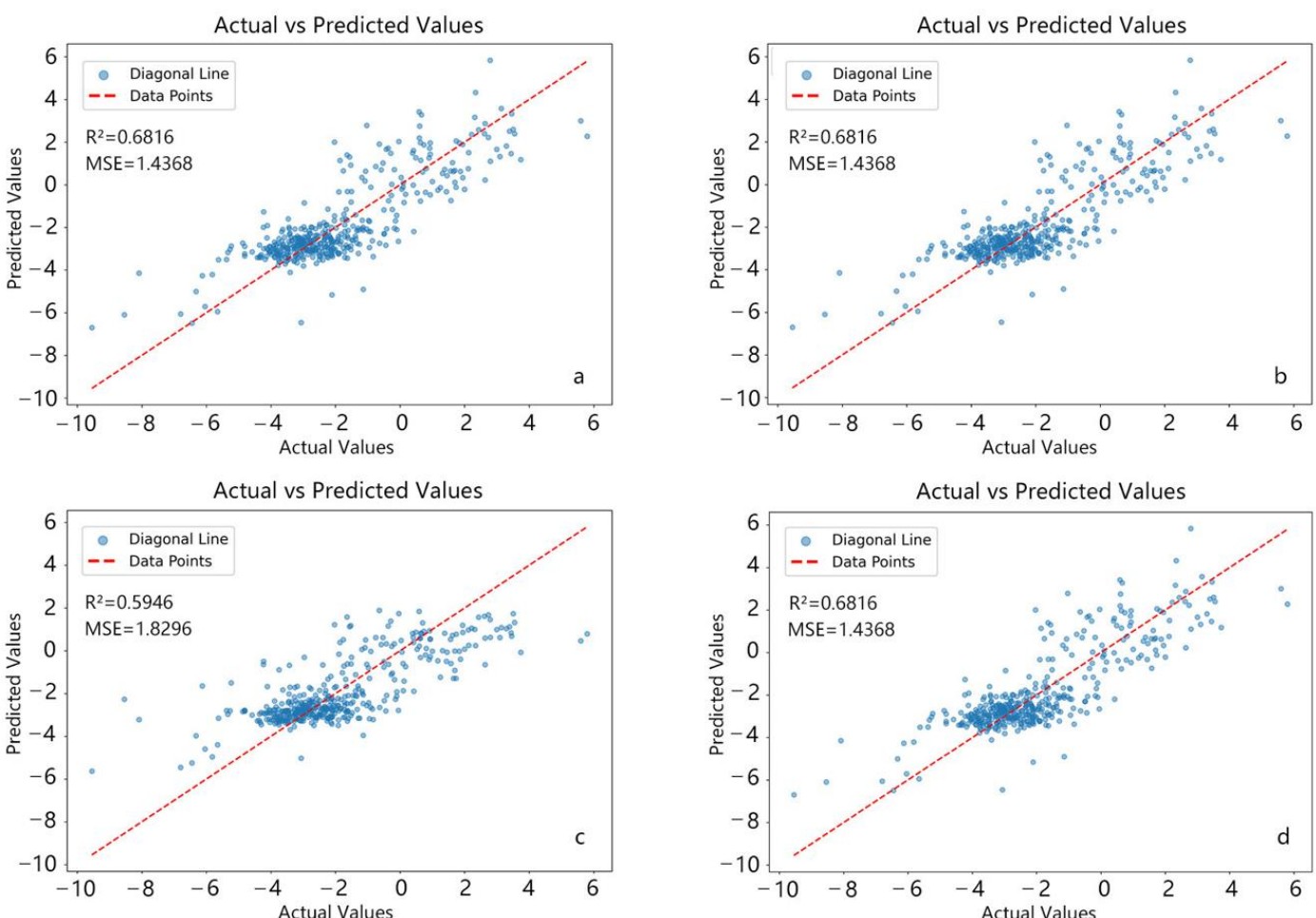

**Figure 3.** Optimal hyperparameter results of the SVR algorithm combined with four hyperparameter-optimization strategies, RS (**a**), GS (**b**), BO (**c**), GA (**d**).

### 3.1.3. Results of the BP Neural Network Algorithm

In this section, we present the results of the BPNN algorithm fine-tuned with the four hyperparameter-optimization strategies. Each hyperparameter-optimization strategy underwent cross-validation to ensure the model's reliability. This step ensures that the hyperparameters we obtained exhibit optimal performance across the entire dataset. Table 4 provides a detailed listing of the critical hyperparameter choices under various optimization strategies, including the units hidden (UH), dropout rate (DR), learning rate (LR), activation (Act), and optimizer (Opt). Concurrently, parameters related to the optimization strategy, such as max trials (MT) and early stopping (ES), were also considered. As observed from Table 4, different hyperparameter-optimization strategies resulted in varied hyperparameter combinations, reflecting each strategy's unique characteristics and preferences when searching within the parameter space.

**Table 4.** Optimal hyperparameter results of the BPNN algorithm combined with four hyperparameter-optimization strategies.

| Optimization Strategy | UH | DR | LR | Act | Opt | MT | ES |
|---|---|---|---|---|---|---|---|
| RS | 128 | 0.1 | 0.004 | ReLU | Adam | 100 | 10 |
| GS | 64 | 0.4 | 0.0078 | ReLU | Adam | 200 | 10 |
| BO | 96 | 0.3 | 0.006 | ReLU | Adam | 200 | 10 |
| GA | 64 | 0.4 | 0.0078 | ReLU | Adam | 200 | 10 |

Several insights emerged after a detailed analysis of the loss function curves under the BPNN algorithm across the four hyperparameter-optimization strategies. For the RS strategy, the curve showed a swift decline in training loss from 3.25 to 2.00 within the initial two iterations, followed by a more gradual decrease. The validation loss was significantly reduced in the first five iterations, but subsequent fluctuations hinted at mild overfitting. Notably, the two loss curves intersected during the 20th, 24th, and 28th iterations, with a loss value ranging from 1.6 to 1.7 (Figure 4a). Under the BO strategy, the training loss curve exhibited a pronounced decline and stabilized after the 8th iteration. The initial decline in validation loss was followed by fluctuations, frequently intersecting with the training curve after the 25th iteration, which suggests a comparable model performance on both the training and validation datasets (Figure 4c). On the other hand, for both the GS and GA strategies, the training loss consistently declined. In contrast, the validation loss remained volatile, rarely intersecting (Figure 4b,d), indicating the relatively weaker performance of models trained under these strategies, with more significant discrepancies. The analysis of the loss function curves shows that different hyperparameter-optimization strategies exhibit varying performance characteristics under the BPNN algorithm. The RS strategy performs well in early iterations but might present mild overfitting later. The BO strategy's model performs similarly on training and validation data, albeit with slight overfitting in some iterations. Conversely, the GS and GA strategies require further optimization, given their relatively weaker model performance and the continued fluctuations observed in their validation loss.

We interpreted the degree of precise alignment between the model predictions and observed values of the BPNN algorithm combined with four different hyperparameter-optimization strategies using the two metrics, $R^2$ and MSE. The RS, BO, and GA approaches displayed similar model performances among these strategies. Their $R^2$ values consistently remained around 0.68, while the MSE ranged between 1.43 and 1.44, which suggests that these three strategies achieved relatively stable and accurate prediction results, as depicted in Figure 4e,h,g. In contrast, with an $R^2$ value of 0.67, the GS strategy was slightly lower than the other three. Its MSE value stood at 1.47, indicating marginally higher prediction errors, as shown in Figure 4f. Upon comparing the predictive performances of the four strategies, it is evident that despite minor discrepancies in specific metrics, they generally offered relatively accurate predictions. Notably, the BO strategy stood out among the four

for its exceptional stability and accuracy in model prediction, making it the most promising predictive strategy in this context.

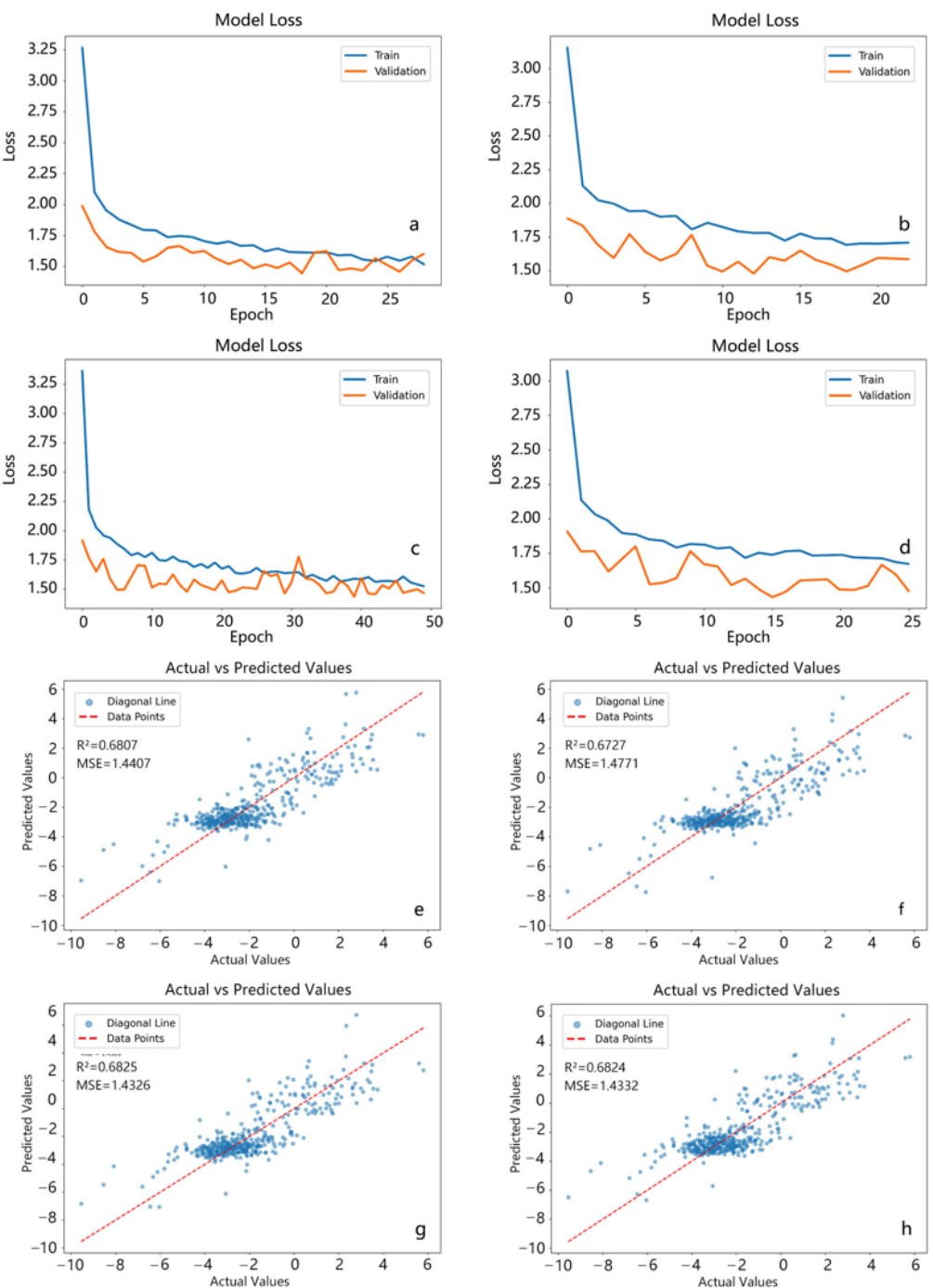

**Figure 4.** BP neural network algorithm training loss function curve: RS (**a**), GS (**b**), BO (**c**), GA (**d**); comparison between predicted values and actual values using the BPNN algorithm, RS (**e**), GS (**f**), BO (**g**), GA (**h**).

### 3.1.4. Results of the Convolutional Neural Network Algorithm

This section presents the optimization outcomes achieved by combining the CNN algorithm with the four hyperparameter-optimization strategies. Cross-validation was also applied to each hyperparameter-optimization strategy to ensure the robustness of the models. Subsequently, we have enumerated the key hyperparameters determined under each optimization strategy (Table 5), namely units hidden (UH), dropout rate (DR),

learning rate (LR), activation (Act), and optimizer (Opt). In addition, parameters directly related to the optimization strategies, such as max trials (MT) and early stopping (ES), were also considered. By observing the optimal hyperparameters, it is evident that each hyperparameter-optimization strategy led to unique hyperparameter combinations, shedding light on each strategy's distinct characteristics and tendencies during their exploration in the parameter space.

**Table 5.** Optimal hyperparameter results of CNN algorithm combined with four hyperparameter-optimization strategies.

| Optimization Strategy | UH | DR | LR | Act | Opt | MT | ES |
|---|---|---|---|---|---|---|---|
| RS | 32 | 0.1 | 0.0038 | ReLU | Adam | 50 | 10 |
| GS | 64 | 0.3 | 0.006 | ReLU | Adam | 200 | 10 |
| BO | 128 | 0.4 | 0.001 | ReLU | Adam | 50 | 10 |
| GA | 32 | 0.3 | 0.0071 | ReLU | Adam | 200 | 10 |

We meticulously examined the loss function curves of models derived from four hyperparameter-optimization strategies under the CNN algorithm. The training and validation loss curves for all four strategies showed gradual convergence. The RS and BO strategies performed better (Figure 5a,c). The validation loss curves for these two strategies sharply dropped from around 2.5 and entered a fluctuating state after the 10th and 5th training epochs, respectively. The validation and training loss curves overlapped multiple times by the end of the 30th epoch for RS and the 25th epoch for BO. The GS strategy's validation loss curve showed more significant fluctuations in the early stages and had difficulty approaching the training loss curve (Figure 5b), indicating potential overfitting or other issues that warrant further analysis. The GA strategy's validation loss curve exhibited continuous fluctuations during the first 40 training epochs. Between the 20th and 30th epochs, the validation loss overlapped with the training loss multiple times, which might suggest a relatively stable learning phase for the model at this stage. However, this does not necessarily indicate good generalization capabilities (Figure 5d). Moreover, the persistent fluctuations might imply overfitting at certain stages of the model's training.

This study used the coefficient of determination $R^2$ and the mean squared error (MSE) to analyze the relationship between the simulated and actual values generated by the model. Upon observing the results from the four strategies, it was evident that the RS and GA strategies displayed closely related performances in model simulations. Their $R^2$ values were 0.70 and 0.69, respectively, with MSE ranging between 1.34 and 1.37, which suggests that these two strategies achieved relatively robust and accurate simulation outcomes (Figure 5e,h). On the other hand, the GS and BO strategies had an $R^2$ value of around 0.67, slightly lower than the former two strategies, and their MSE reached 1.46, indicating a slightly higher error in simulations for these two strategies (Figure 5f,g). When considering the simulation effects of all four strategies, it is notable that despite minor differences in specific evaluation metrics, they all provided relatively accurate simulation results, especially the RS strategy.

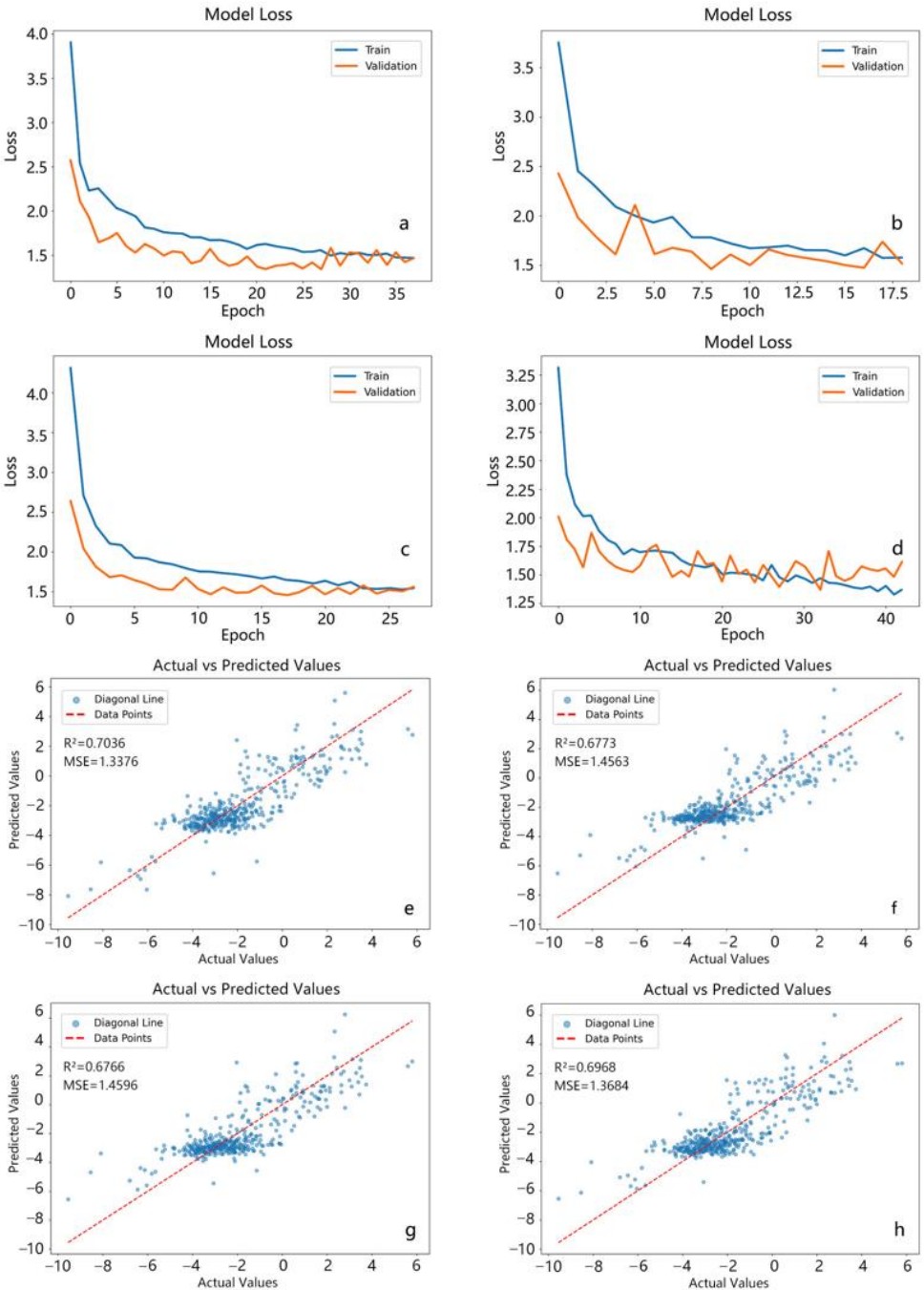

**Figure 5.** CNN algorithm training loss function curve: RS (**a**), GS (**b**), BO (**c**), GA (**d**); comparison between predicted values and actual values using the CNN algorithm, RS (**e**), GS (**f**), BO (**g**), GA (**h**).

### 3.2. Selection of the Optimal Prediction Model for Southeast Asia's NEP

In this section, we compared the predictive results obtained by integrating the 4 machine-learning algorithms with the 4 hyperparameter-optimization strategies, which yielded 16 models. By inputting Southeast Asia's time-series remote sensing feature variables, we obtained the annual NEP of Southeast Asia from 2001 to 2020 for each model. For our comparative analysis, we specifically looked at the NEP of Southeast Asia in 2010. Models based on the RF algorithm combined with the four hyperparameter-optimization strategies showed consistent results (Figure 6a–d). They adequately represented the vegetation spatial differentiation in Southeast Asia. Conversely, models derived from the SVR algorithm in conjunction with the four hyperparameter-optimization strategies struggled

to accurately represent the actual regional NEP (Figure 6e–h), indicating that the SVR algorithm's generalization capability for this specific problem might be limited. The BPNN algorithm combined with the four optimization strategies demonstrated promising results, capturing the influence of factors such as land cover, topography, and human activity on the NEP (Figure 6i–l). While the CNN algorithm, combined with the four optimization strategies, had spatial results broadly consistent with those from the BPNN algorithm, there were discrepancies. Specifically, the CNN results indicated that the NEP in agricultural areas (like the rice fields in Thailand's central plains) was greater than zero (Figure 6m–p), which contradicts our understanding. In conclusion, among the analyzed algorithms, the RF and BPNN algorithms proved to be the most proficient in simulating the NEP of Southeast Asia.

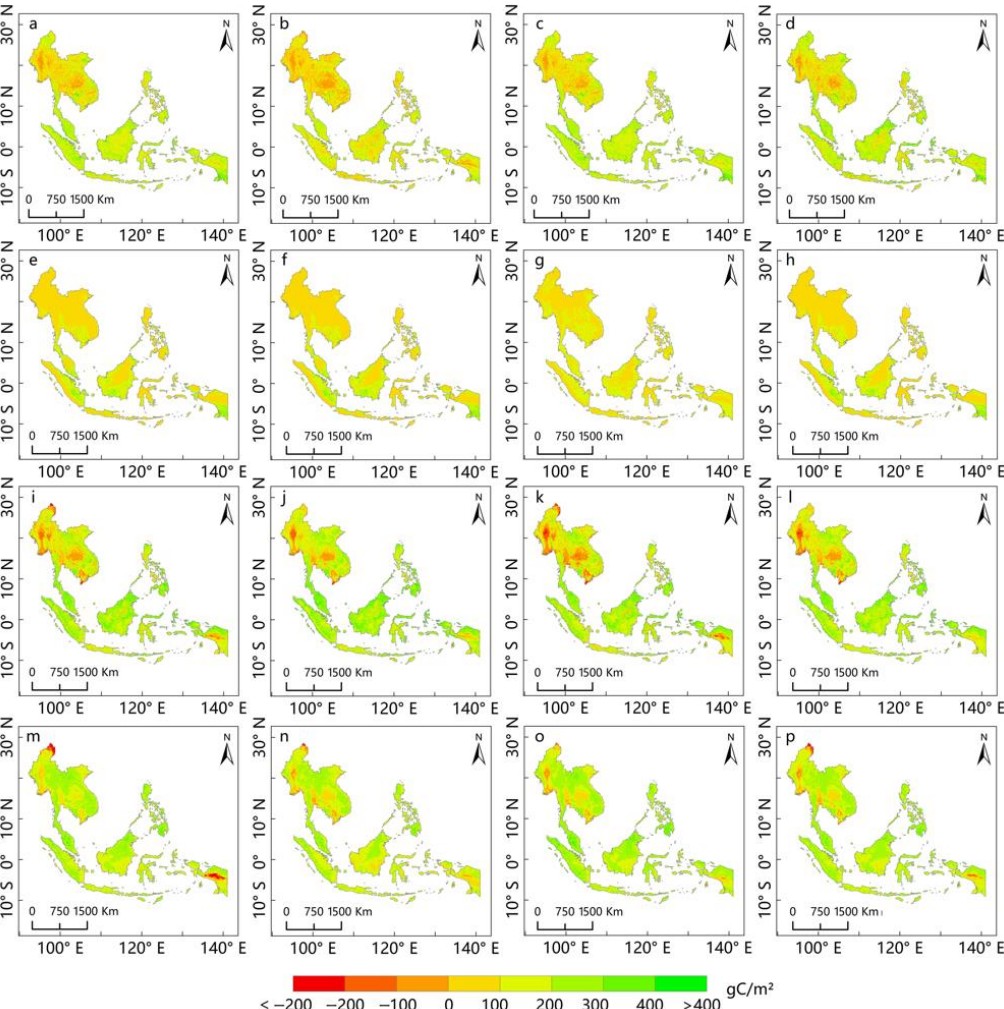

**Figure 6.** Predicted NEP results for Southeast Asia (2010) using four machine-learning algorithms in combination with four hyperparameter-optimization strategies: the RF algorithm with the RS strategy (**a**), RF algorithm with the GS strategy (**b**), RF algorithm with the BO strategy (**c**), RF algorithm with the GA strategy (**d**), SVR algorithm with the RS strategy (**e**), SVR algorithm with the GS strategy (**f**), SVR algorithm with the BO strategy (**g**), SVR algorithm with the GA strategy (**h**), BPNN algorithm with the RS strategy (**i**), BPNN algorithm with the GS strategy (**j**), BPNN algorithm with the BO strategy (**k**), BPNN algorithm with the GA strategy (**l**), CNN algorithm with the RS strategy (**m**), CNN algorithm with the GS strategy (**n**), CNN algorithm with the BO strategy (**o**), CNN algorithm with the GA strategy (**p**).

Based on our detailed analysis of models generated by pairing each machine-learning algorithm with the four hyperparameter-optimization strategies, we identified the best-

performing combinations for each algorithm: RF with RS, SVR with both GS and RS (or GA), BPNN with BO, and CNN with RS. We then employed these four optimal models to predict the monthly NEP for Southeast Asia (Figure 7). Notably, the models resulting from the RF algorithm combined with the RS strategy, the BPNN algorithm combined with the BO strategy, and the CNN algorithm combined with the RS strategy produced NEP estimates that were closely aligned. In contrast, the NEP predictions from the SVR algorithm (when paired with the GS, RS, or GA strategies) were considerably lower than the other three models, which exhibited a more chaotic monthly pattern throughout the year. Additionally, the RF algorithm combined with the RS strategy showed a significant disparity between the highest and lowest NEP values throughout the year. Since a large part of Southeast Asia comprises tropical rainforests, we expect relatively consistent NEP values throughout the year. However, the model's output contradicts this expectation. Considering the performance of all 16 models and their capabilities in simulating the NEP of Southeast Asia, we are inclined to believe that the combination of the BPNN algorithm with the BO strategy offers the most reliable and accurate results.

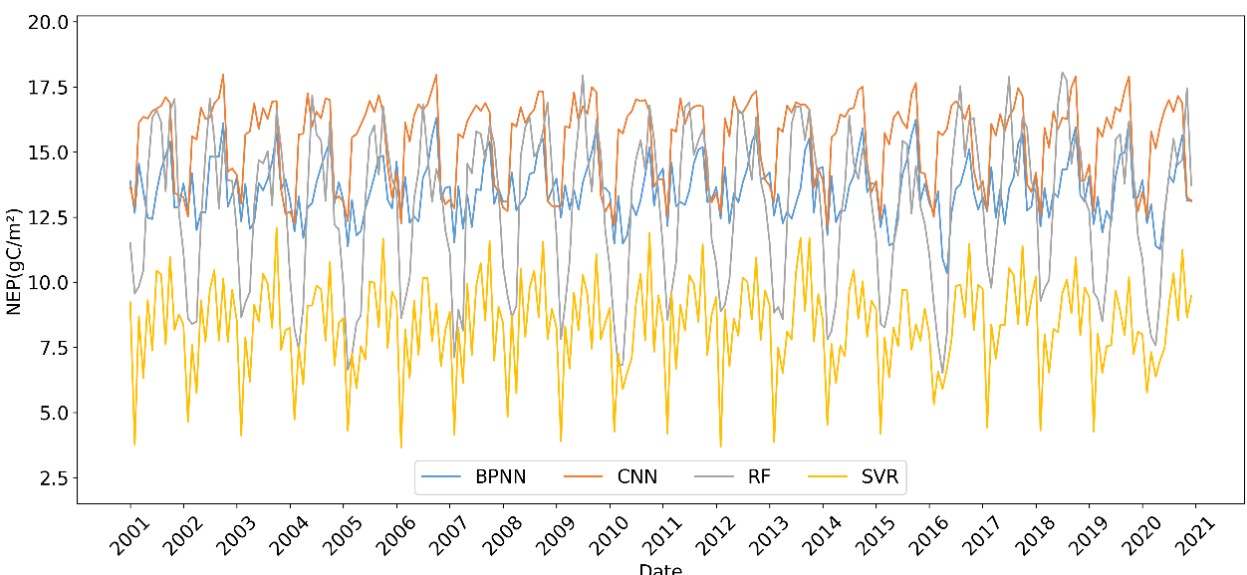

**Figure 7.** Monthly predicted NEP of Southeast Asia using the four optimal machine-learning algorithms in combination with hyperparameter-optimization strategies. The BPNN algorithm with the BO strategy, CNN algorithm with the RS strategy, RF algorithm with the RS strategy, and SVR algorithm with the GS strategy.

### 3.3. Validation of the Rationality of the Optimal Prediction Model for NEP in Southeast Asia

The model predictions based on the BPNN algorithm and the BO strategy for NEP at a regional scale align closely with the results from the GEODA and NIES products. The annual NEP size strongly correlates with land cover and ecosystem types. Generally, forest ecosystems act as carbon sinks, grassland ecosystems lie between carbon sinks and carbon sources, while farmland ecosystems predominantly serve as carbon sources. Specifically, areas like the tropical rainforests of the Indochinese Peninsula and the Malay Archipelago demonstrate higher NEP values, with multi-year NEP values often exceeding $300 \ gC/m^2$. In contrast, natural shrublands, grasslands, and plantation ecosystems show multi-year NEP values typically ranging from 0 to $200 \ gC/m^2$. Farmland ecosystems predominantly exhibit multi-year NEP values between $-100$ and $0 \ gC/m^2$ (Figure 8a–c). The NEP data for Southeast Asia provided by GEODA indicate that forest ecosystems largely have NEP values above $400 \ gC/m^2$. Natural shrublands, natural grasslands, and plantation ecosystems have multi-year NEP values ranging from 0 to $300 \ gC/m^2$, while farmland ecosystems show multi-year NEP values essentially between $-200$ and $0 \ gC/m^2$

(Figure 8d–f). On the other hand, NIES's NEP data also demonstrate forest ecosystem NEP values consistently above 400 gC/m$^2$, with some reaching considerably higher values. Natural shrublands, grasslands, and plantation ecosystems exhibit multi-year NEP values between 0 and 400 gC/m$^2$. Farmland ecosystems generally show multi-year NEP values between -200 and 0 gC/m$^2$, with some areas dropping below $-200$ gC/m$^2$ (Figure 8g–i).

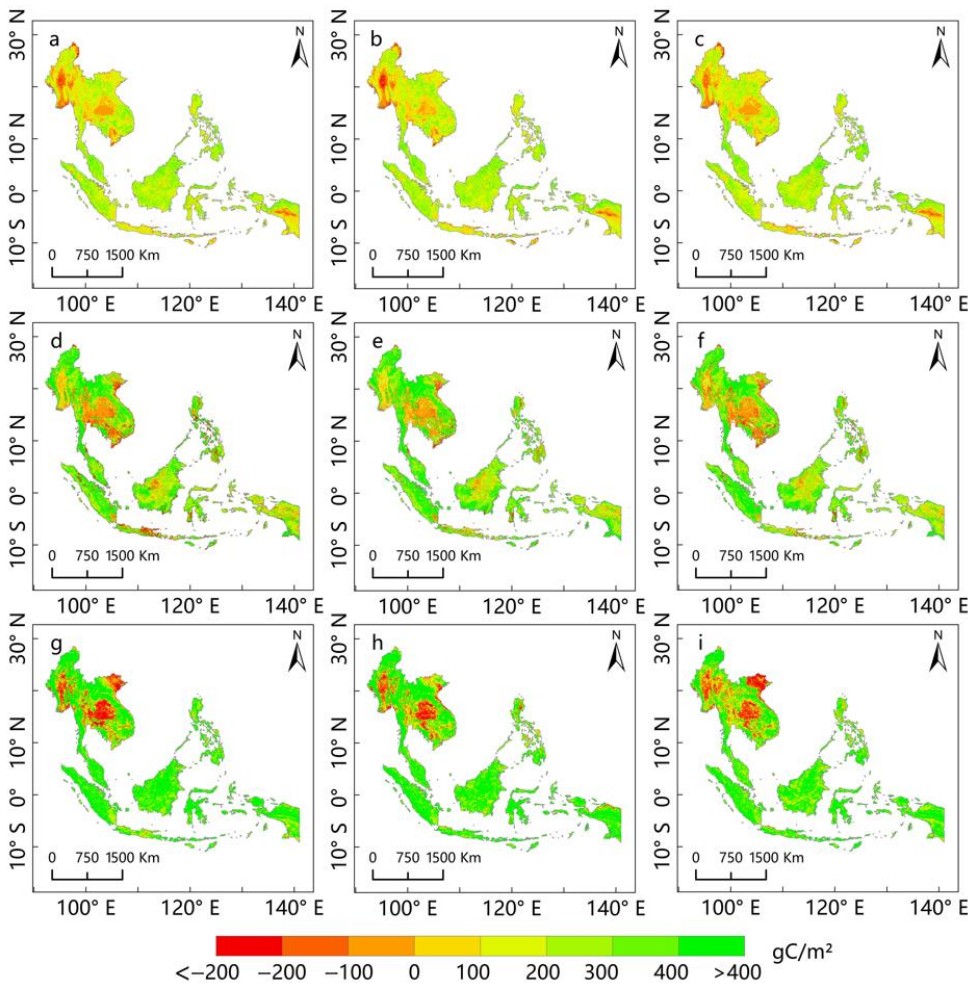

**Figure 8.** Comparison of the NEP results predicted by the BPNN algorithm combined with the BO strategy, with GEODA and NIES NEP data products. The BPNN algorithm combined with the BO strategy predicted NEP in Southeast Asia for the years 2001 (**a**), 2009 (**b**), and 2018 (**c**). GEODA's NEP data product for Southeast Asia for the years 2001 (**d**), 2009 (**e**), and 2018 (**f**). NIES's NEP data product for Southeast Asia for the years 2001 (**g**), 2009 (**h**), and 2018 (**i**).

We compared our predictions year-by-year, based on the BPNN algorithm combined with the BO strategy, with the annual NEP values provided by NIES and GEODA. As predicted by our model, the average yearly NEP for Southeast Asia stood at 162.49 gC/m$^2$. The highest value was registered in 2012 at 166.48 gC/m$^2$, while the lowest was in 2016, recording 156.17 gC/m$^2$. In the same timeframe, the NIES NEP product had an average annual value of 399.24 gC/m$^2$. Its peak value was in 2008, reaching 455.12 gC/m$^2$, and its lowest was in 2018, dropping to 352.54 gC/m$^2$. Meanwhile, the average NEP for Southeast Asia from the GEODA product during the same period was 268.43 gC/m$^2$. The highest value for GEODA was observed in 2014, at 290.03 gC/m$^2$, while its lowest came in 2010, with a value of 255.37 gC/m$^2$. All three sets of results or data products suggest a gradual decline in the NEP for Southeast Asia over the past two decades (Figure 9).

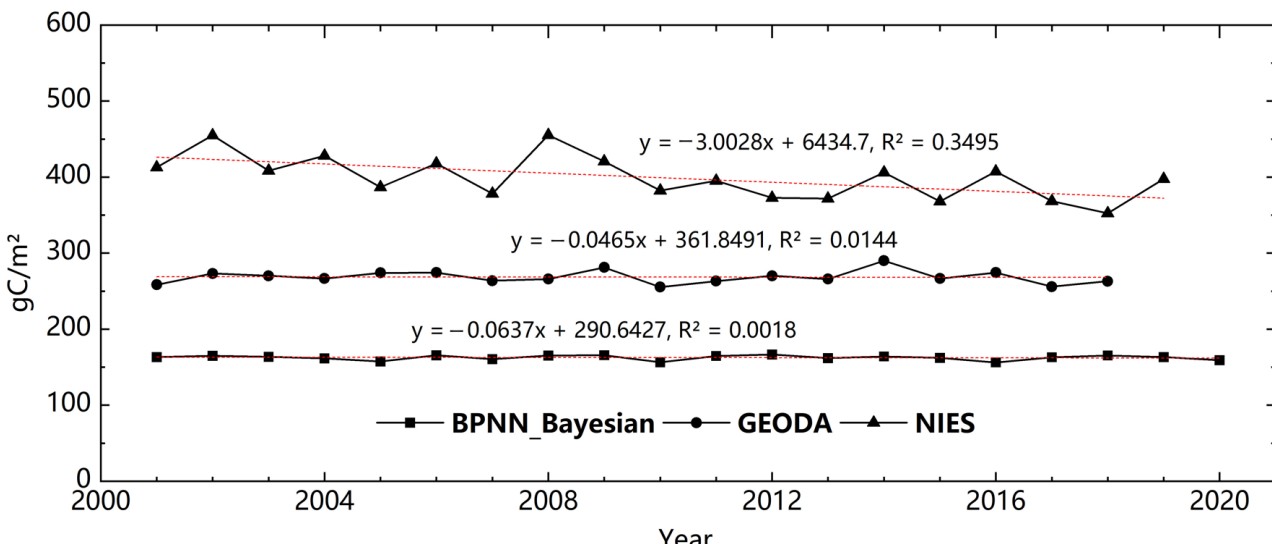

**Figure 9.** Comparison of the NEP predicted by the BPNN algorithm combined with the BO strategy for Southeast Asia from 2001 to 2020 with GEODA and NIES NEP products.

Every NEP prediction exhibits disparities due to differences in model algorithms and data sources. We conducted a correlation analysis between our optimal model's multi-year NEP predictions for Southeast Asia and the NEP data products from GEODA and NIES. We analyzed these three spatial predictive outcomes pairwise and found a relatively high correlation between the optimal model's prediction and the NIES results. Most areas in the Indochina Peninsula, Malay Peninsula, Kalimantan Island, and the Philippine Archipelago demonstrated a moderate-to-strong correlation (Figure 10a). However, the spatial correlation of the optimal model's NEP prediction with the GEODA data product was slightly weaker than the former, with several regions across the entire domain showing a high degree of uncertainty (Figure 10b).

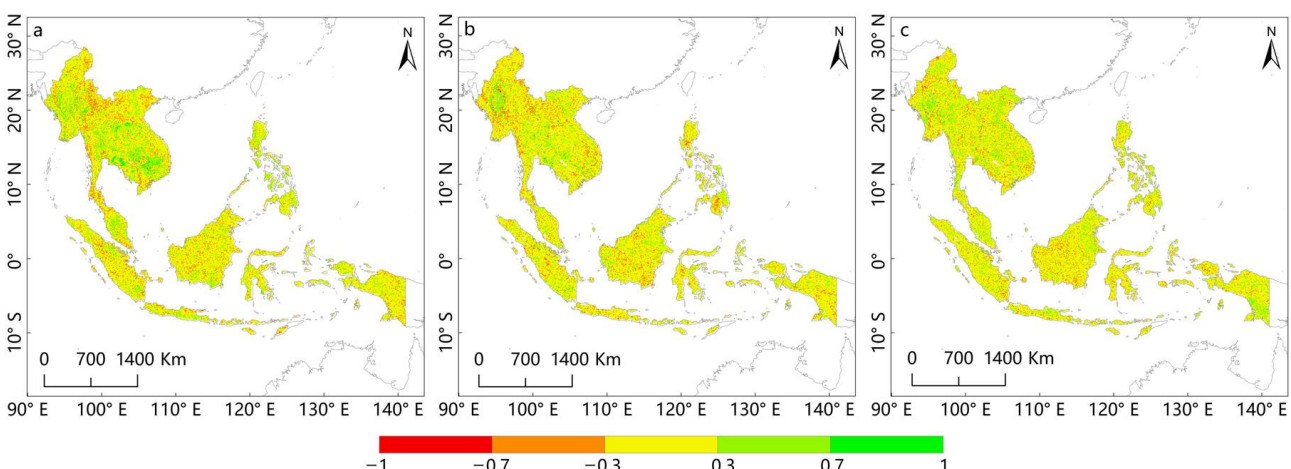

**Figure 10.** Correlation analysis comparison of three NEP prediction results in Southeast Asia from 2001 to 2020. Correlation between the prediction results of the BPNN algorithm combined with the BO strategy and the NIES product (**a**), correlation between the prediction results of the BPNN algorithm combined with the BO strategy and the GEODA product (**b**), and correlation between GEODA and NIES data products (**c**).

The NEP predictions for Southeast Asia by NIES and GEODA also exhibited considerable spatial uncertainties (Figure 10c). Such disparities might originate from variations in data consistency, methodologies, simulation scales, or perhaps optimizations in estima-

tion methods, including the selection of algorithms and hyperparameter tuning [24,27]. These elements underscore the intrinsic uncertainties in regional-scale NEP predictions.

## 4. Discussion

### 4.1. Comparison of the Performance of Different Machine-Learning Algorithms

This study compared the model-fitting effects of four machine-learning algorithms, each combined with different hyperparameter-optimization strategies. Judging by the performance metrics $R^2$ and MSE, deep-learning algorithms like BPNN and CNN generally outperformed the results of RF and SVR algorithms. Notably, the model fitted using CNN combined with the random search strategy performed exceptionally well, with an $R^2$ of 0.7036 and an MSE of 1.3376. However, a significant shift was observed when we applied these models to actual feature target inversion. We deeply analyzed and interpreted the gridded results of the model-inverted feature targets, especially considering the spatial representation of NEP in different ecosystems, such as the carbon balance characteristics and spatial distribution trends in forests, grasslands, shrubs, and farmlands. Theoretically, original tropical forests, shrubs, and wild grasslands in Southeast Asia should act as carbon sinks, i.e., NEP > 0 [51–53]. Given the extensive farmland in this region, especially vast paddy fields emitting substantial greenhouse gases during their growth, and considering factors like field management and land conversion under the premise of climate change, many scientific studies have deemed the farmlands in this region as carbon sources, i.e., NEP < 0 [54,55]. Surprisingly, the model fitted using the CNN algorithm predicted NEP values in farmlands that were almost all greater than 0, with most above 20 gC/m$^2$, which significantly differs from theoretical expectations and results from similar data products. In contrast, the model fitted using the BPNN algorithm combined with the Bayesian optimization strategy demonstrated a robust performance in terms of $R^2$ and MSE but, more importantly, could reflect the spatial patterns of NEP in different ecosystems more accurately. Additionally, its spatial distribution is closely aligned with international products of a similar kind. After weighing both the predictive accuracy of the model and its practical application effects, we ultimately chose the model fitted using the BPNN algorithm combined with the BO strategy as the most suitable model.

Comparing the simulation results of the BPNN and CNN algorithms, we found that the former is more consistent with reality, especially in farmland ecosystems like rice paddies. We also analyzed potential factors that might lead to discrepancies in the simulation results of these two algorithms. Firstly, the difference in the structural design of the algorithms could be a reason. CNNs, with their convolutional layers, are designed to capture local patterns in the input data. In contrast, BPNN, being a fully connected network, aims to learn features from a global perspective. Given this difference, these two structures might exhibit distinct behaviors when processing the same dataset. Another influencing factor could be the activation function. Both algorithms employ the ReLU activation function, which behaves differently over various intervals. During different training phases, if the initialization or updates of weights for specific neurons result in negative outputs, ReLU would set them to zero, leading to some parts of the model becoming "inactive" during training. Due to the structural differences, CNNs and BPNNs might react differently in such scenarios. Furthermore, the nature of the data itself might also contribute to the observed discrepancies. CNNs are primarily designed to handle static data, like images. At the same time, BPNNs, owing to their fully connected nature, might be better suited to capture the characteristics of temporal data, thereby reflecting the actual scenario more accurately.

### 4.2. Comparison of Multiple Hyperparameter-Optimization Methods

The selection of hyperparameters plays a pivotal role in the performance of machine-learning models. To ensure the optimal performance of each algorithm, we employed four distinct hyperparameter-optimization strategies: random search (RS), grid search (GS), Bayesian optimization (BO), and genetic algorithm (GA). Each strategy possesses unique strengths. However, the combination of BPNN and BO exhibited a superior performance

in this study. Bayesian optimization is a global optimization method grounded in probabilistic models. It adeptly selects new parameter combinations in every iteration based on existing results. Compared to traditional GS and RS methods, BO is more efficient as it harnesses accumulated knowledge from prior iterations to guide subsequent searches, thus averting redundant computations. Additionally, BO strikes an impeccable balance between exploration and exploitation, which means it can venture into new regions of the parameter space while leveraging known information to enhance the model performance [56,57].

In this study, the model combining BPNN with the BO strategy demonstrated an exceptional performance in both R-squared and MSE metrics, achieving an R-squared of 0.6825 and an MSE of just 1.4326. This outcome significantly surpasses the results obtained from the other three hyperparameter-optimization strategies. Even more crucially, when applying this model to the inversion of feature variable grid data, it vividly depicts the carbon balance characteristics and spatial distribution trends of different ecosystems, aligning closely with international counterparts. Consequently, the model integrating BPNN and BO outshined all of our experiments. It showcased an exemplary performance during the model training process and in practical applications.

### 4.3. Integrating Machine Learning and Ecological Process Models for a New Perspective in NEP Prediction

Models built using machine-learning algorithms often have many parameters and intricate structures, which require substantial data and computational resources for their training. Achieving an optimal model performance necessitates hyperparameter tuning, which typically involves multiple rounds of model training and validation, further compounding computational intricacy. However, their forward propagation (i.e., predictions) is typically swift once these models are adequately trained [58,59]. Vegetation ecological process models, on the other hand, are rooted in explicit physiological and environmental processes, typically involving a set of differential equations. Although these models may appear structurally more straightforward, their parametrization and calibration can be intricate, requiring vast amounts of field observation data [60,61]. Moreover, as these models are often time-stepped, they may necessitate extended durations for simulating long time-series data. From a computational complexity standpoint, machine-learning models demand more resources during the training phase but are quicker in the prediction phase. In contrast, vegetation ecological process models might be more time-consuming during simulations, especially for extended time-series data.

Machine-learning models are data-driven, implying they do not necessarily adhere to ecological and biophysical principles, leading to the models producing non-physical or non-ecological predictions in certain scenarios. For instance, a machine-learning model might discern relationships between certain features and NEP without any ecological justification. Conversely, vegetation ecological process models are constructed based on a profound understanding of plant growth, photosynthesis, and respiration processes [62,63]. Thus, their predictions typically have clear ecological and biophysical interpretations, which grants these models an advantage in explaining and understanding ecosystem processes. From the standpoint of computational principle validity, vegetation ecological process models, underpinned by explicit ecological and biophysical foundations, excel in interpreting ecosystem processes. While machine-learning models can discern intricate relationships from data, they might lack ecological or plant growth principle interpretations for these relationships [64,65].

Our study demonstrates the unique advantages of machine-learning models in the annual prediction of NEP in Southeast Asia. Often, the choice of method and model for predicting regional-scale NEP depends on the research objectives, available data resources, and interpretability requirements of the model. We believe that, based on the strengths and weaknesses of machine-learning algorithm models and vegetation ecological process models, constructing an innovative integrated model will be the direction of subsequent research. Such an integrated model would not only emphasize the precision of machine-

learning algorithms in data-driven prediction but also incorporate the core characteristics of vegetation ecological process mechanisms, offering a novel perspective for NEP prediction.

## 5. Conclusions

In conclusion, our study advances the field of ecological modeling by demonstrating the effectiveness of integrating advanced machine-learning algorithms with hyperparameter-optimization strategies to enhance the prediction of NEP in Southeast Asia. Specifically, the BPNN algorithm, when fine-tuned with BO, emerged as a powerful tool, reinforcing the value of sophisticated computational techniques in ecological research. Moreover, this work provides a methodological foundation for incorporating machine learning into traditional ecosystem analysis, pointing towards a new horizon in predictive accuracy and reliability. The promising results from deep-learning network models, particularly in complex data-driven contexts, advocate for a paradigm shift from conventional models to more agile, robust, and nuanced analytical frameworks. This study suggests a roadmap for future research to explore the synergy of data-driven and process-based models. Such integration has the potential to unlock deeper insights into ecosystem dynamics, offering a comprehensive toolkit for scientists to monitor, predict, and manage the ecological balance in response to environmental changes. Our findings underline the importance of technological innovation in environmental science and pave the way for the development of more sophisticated, accurate, and scalable models for NEP prediction.

**Author Contributions:** All authors contributed extensively to the study presented in this manuscript. C.H. (Chaoqing Huang), S.H. and C.H. (Chao He): conceptualization, methodology, data curation, visualization, writing the original draft. C.H. (Chaoqing Huang), S.H., B.C., Y.W., P.T. and S.W.: editing, supervision. C.H. (Chaoqing Huang), J.Z., C.H. (Chao He), H.Y., M.N. and C.S.: data curation, investigation, validation. All authors have read and agreed to the published version of the manuscript.

**Funding:** This research received no external funding.

**Data Availability Statement:** The data used to support the results of this research are shown in the manuscript and available from the corresponding author upon request.

**Conflicts of Interest:** The authors declare that they have no known competing financial interests or personal relationships that could have appeared to influence the work reported in this paper.

## Appendix A. Breakdown of the NEP Prediction Workflow

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
