# Peer review of "Synergistic Application of Multiple Machine Learning Algorithms and Hyperparameter Optimization Strategies for Net Ecosystem Productivity Prediction in Southeast Asia"

_remotesensing, doi:10.3390/rs16010017_

Round 1

Reviewer 1 Report

Comments and Suggestions for Authors

Concluding statements though good but a bit contradictory though presented as an improvement over the other or the need to integrate both. Can the concluding narrative be improvised as per the objective of the paper.

As an example, I am referring to concluding narrative such as: 'As compared to traditional ecological process models, data-driven machine learning models have advantages in computational efficiency and result reliability. However, traditional models are more convincing in terms of theoretical soundness'.

Comments on the Quality of English Language

Minor improvements are desirable. As examples, line 412-413 We detailedly analyzed the loss function curves under the CNN algorithm for the four hyperparameter optimization strategies ; Another one, Line 102-103 Based on the abovementioned, this study primarily leverages continuous observational data from the FLUXNET stations in the ASEAN region... etc.

Author Response

Dear Reviewer,

Thank you for your valuable comments and suggestions. Due to the detailed nature of our responses, we have prepared a comprehensive reply in the attached document for your review. We appreciate your time and consideration in reviewing our manuscript.

Best regards,
Dr. Chaoqing Huang

Reviewer 2 Report

Comments and Suggestions for Authors

Dear Authors;

Your work is really very impressive, I read it with great curiosity to evaluate it. I just want to write a few words for your correction, although it is not very necessary; I think it would be a more accurate decision if you move the advantages and disadvantages of NEP, which you gave in the introduction of your article, to the discussion part of your article.

Author Response

(The authors gave the same response as above.)

Reviewer 3 Report

Comments and Suggestions for Authors

Accurate prediction of NEP through machine learning algorithms has great importance in ecological research. However, there are still some mistakes in this study.

1. Please outline the enhanced importance of your method at the ending of Abstract section.

2. Insert essential references at section 4.3.

3.

Comments on the Quality of English Language

It is clear in this version.

Author Response

(The authors gave the same response as above.)

Reviewer 4 Report

Comments and Suggestions for Authors

Title: ASEAN need to be explained what does it means. Replace for “in Southeast Asia”.

Abstract: Line 25 must be more broad change for “in many ecological processes and provision of ecosystem services”. Remove climate change... it is a good proxy but in combination with other variables.  Before accurate add “In example,”. You need to add some sentences about methods and statistics.

Keywords: ASEAN are not a good one, it not self-explains. These words must be different from title.

Introduction: One paragraph about the importance of NEP is enough. Reduce this topic in the introduction. Focus on those topics that you study in the paper, e.g. methodology. ASEAN what it means? Must be explained the first time (line 103).

Lines 102 to 117 are poorly written. This is a mix of descriptive activities and methods, and justifications of the study. Remove this paragraph. Start with one GOOD and clear objective linked to the title. Add some hypothesis or questions to be solved during this research. Remove methods or results. These questions must order the following sections (methods, results, discussion).

Methods: When you read the objective and questions, you can imagine the methods... Here, I found a lot of methods applied... and I have no idea why... so, the last paragraph of the introduction is crucial. Figure 2 can be moved to the Annex.

Results: There are a lot of results here answering nothing related to the objective. They look like a technical report. There are a lot of information, but not related to a clear history... you need to clarify what do you want to do... not just ADD all the information you have. You need to present JUST what do you need to answer specific questions. I wonder if you want to compare different methodologies, and just showed these comparisons... or you want to present a NEW methodology? I am not sure in your draft. So, you need to improve the objective and questions, and then, answer in the results these questions... and remove those results that not answer anything.

Reduce the figures and tables to the minimum necessary.

Discussion must be checked after these changes.

Comments on the Quality of English Language

It is good English, minor changes are required, some paragraph must be ordered to clarified the contents.

Author Response

(The authors gave the same response as above.)

Reviewer 5 Report

Comments and Suggestions for Authors

The authors employed four machine learning algorithms (RF, SVR, BPNN, and CNN) in conjunction with four hyperparameter optimization strategies (RS, GS, BO, and GA) to predict the Net Ecosystem Productivity in the ASEAN region. By comparing 16 combination methods, they identified the optimal predictive model for NEP in the ASEAN region. The results were further analyzed and compared with findings from international peers in the field. Overall, the paper has done lots of analyses and works that are generally valuable, but there are still a few suggestions for the authors to consider whether to accept or not.

Comments:

1. Lines135~137 need to add relevant references to the introduction of NEE.

2. The entire manuscript should improve the quality of the figures.

3. Lines167~168 need to indicate how many indicators have been utilized.

4. Readjust the flow chart in Figure 2. The process and legend should be distinguished, otherwise, it will easily confuse. For example, you can place the legend on one side and add a label, or use different colors to distinguish the legend.

5. The meanings of sub-figures a, b, c, and d in Figure 4 should be described properly in the caption.

6. Line 313: In Figure 3d, the result of R2 is 0.2241, which is quite different from the results of other models. How do you explain it?

7. Please explain why Figure 9 compares the NEP data of 2001, 2009, and 2018, Why choose these three years?

Author Response

(The authors gave the same response as above.)

Round 2

Reviewer 4 Report

Comments and Suggestions for Authors

Thanks for improving the manuscript. Now, the objective is clear. However, the methods still need to be better linked to objectives, e.g. why you conduct each analysis in order to answer each objective. If you explain this in methods, it much be better. The same for objectives... here you present a LOT of tools... but it is no clear HOW they answer each question. It can be better if you say "in order to answer Objective X, we made BLA BLA". To many results for few questions to answer.

Finally, the Conclusions still must be improved. To date is a result summary... and conclusions are the NEW knowledge that you made for Science... it is more general and broad than your results... Actually, teh can be named as "Final Remarks". Please, improve this section in the final draft.
